# Climate change belief systems across political groups in the United States

**Sanguk Lee**[1]*, **Matthew H. Goldberg**[1], **Seth A. Rosenthal**[1], **Edward W. Maibach**[2], **John E. Kotcher**[2], **Anthony Leiserowitz**[1]

1 Yale Program on Climate Change Communication, Yale University, New Haven, Connecticut, United States of America, 2 Center for Climate Change Communication, George Mason University, Fairfax, Virginia, United States of America

* lswook555@gmail.com

**Data Availability Statement:** All files including data, R code, R code's output, and a minimal dataset are available from OSF (URL: https://osf.io/4e5sh/, DOI: 10.17605/OSF.IO/4E5SH).

## Abstract

Beliefs and attitudes form the core of public opinion about climate change. Network analysis can reveal the structural configuration of these beliefs and attitudes. In this research, we utilize a belief system framework to identify key psychological elements, track change in the density of these belief systems over time and across political groups, and analyze the structural heterogeneity of belief systems within and between political groups in the United States. Drawing on fifteen waves of nationally representative survey data from 2010 to 2021 ($N = 16,742$), our findings indicate that worry about climate change is the most central psychological element. Interestingly, we find that among politically unaffiliated individuals, the connections between psychological elements have strengthened over time, implying an increase in the consistency of belief systems within this group. Despite the political polarization in beliefs about climate change between Republicans and Democrats, our findings reveal that the ways these two groups organize and structure climate change beliefs systems are not markedly different compared to those of other groups. These findings provide theoretical and practical insights for climate change experts and communicators.

## Introduction

The successful implementation of many climate solutions hinges on public approval and adoption. As such, understanding public beliefs and attitudes about climate change is critical. Academics often examine the relationships among diverse psychological elements, such as beliefs, risk perceptions, and policy support. Earlier research has found that beliefs that climate change is real and human-caused, perception of scientific consensus, and risk perceptions about climate change are associated with support for climate actions and policies [1–3].

This literature offers valuable insights into public beliefs and attitudes about climate change. However, it has historically focused on relationships between specific psychological elements without fully addressing the interconnected nature of these components within a broader belief system. For example, although worry and risk perceptions are crucial predictors of climate policy support [1], our understanding of how these psychological elements are

**Funding:** The author(s) received no specific funding for this work.

**Competing interests:** The authors have declared that no competing interests exist.

structurally embedded within people's broader system of climate-related beliefs remains limited. A prior study conducted in 22 European countries has shed some light on this question, revealing that worry about climate change stands as a central component strongly connected with its adjacent psychological elements within these systems [4]. However, whether these findings are applicable to the U.S. context, where climate change is highly politicized, remains unclear.

The U.S. is a major emitter of carbon dioxide [5], underscoring the importance of examining the structure and organization of public belief systems regarding climate change. Given the strong political polarization of the issue in the U.S., an analysis of how these belief systems vary among different political groups, is also important. Further, it is particularly pertinent to study the evolution of these belief systems over the past decade, a period during which there has been a notable positive change in climate change perceptions in the U.S [6]. By examining these changes from a structural perspective, we can gain insights into the effectiveness of communication efforts. Such an analysis can also reveal whether these collective efforts have helped to shape a more cohesive and coherent belief system regarding climate change in the American context.

In the current study, a climate change belief system is defined as a network of interrelated beliefs, attitudes, and behaviors relevant to climate change [7, 8]. Within this framework, nodes symbolize beliefs (and/or attitudes and behaviors) and edges depict the relationships between them. A belief system is empirically constructed by interlinking these individual elements based on their associations. Although behavior is not typically classified as an integral component of belief systems in a direct sense, it is certainly associated with beliefs, attitudes, and perceptions. As belief systems guide behaviors and, reciprocally, behaviors can affirm or contradict beliefs, we argue that this action-reflection cycle forms a key part of belief systems. Hence, we integrate both psychological and behavioral elements into our investigation of climate change belief systems.

Recent advances in network analysis have helped researchers estimate the *structural properties* of a complex psychological system where an array of psychological elements are interconnected. This study examines two key structural dimensions: centrality and density. Centrality, based on a structural position of a psychological element within the belief system, enables the identification of particularly influential elements [7, 9]. Mapping a belief system allows for the estimation of the structural position of each psychological element, such as whether a particular belief is central or peripheral within a network. Elements centrally located in a belief system typically carry more weight than peripheral ones, due to their potentially stronger influence on other beliefs within the system [10]. For example, if worry about climate change is the central node in a network, changes in the level of worry would subsequently be expected to have a larger cascading impact on the rest of the network than a similar level of influence on a less-central element. Network analysis can help develop more effective climate change communication strategies by identifying which psychological elements are likely to be the most influential within a belief system.

Density, representing the overall strength of the connections among elements [11], also provides valuable insights for climate change communication. A dense belief system denotes high correlations among its constituent elements. Individuals with a dense belief system exhibit less randomness and disorder in their opinions, and exhibit more consistency and stability in their psychological processes [8, 9, 12, 13]. However, given that elements within a dense belief system are tightly interwoven, influencing the beliefs and attitudes of individuals with such a system can be more difficult [12]. Conversely, individuals with a less dense network are more likely to exhibit randomness and disorder in their opinions [13], with less coherent views on subjects such as climate change. Nevertheless, strategic communication can

have a greater impact on them, as there is more opportunity to insert new belief elements and/ or strengthen existing connections between elements. Network density analysis can then help communicators optimally allocate communication resources. Providing structured information about a topic to those with less organized belief systems can increase their belief system density, while individuals with high-density systems may prove less receptive.

The density of belief systems has been examined in political contexts. Scholars have found that politically informed individuals tend to have denser political belief systems than do politically apathetic individuals [8, 14, 15]. This could potentially explain why political elites consistently demonstrate relatively well-defined and coherent opinions on a range of political issues [8, 9]. A similar pattern can be observed with climate change belief systems, especially in the U.S., where climate change opinions have become deeply intertwined with political ideologies [16, 17]. As a result, individuals with strong political affiliations are likely to have denser climate change belief systems than those who are politically unaffiliated.

The structure of a network shapes the flow of information within it. Depending on this structure, a shift in a single belief could either rapidly or only gradually propagate, influencing either the entire system or just a portion of it. Comparing structures within and between political groups can also illuminate the organization of people's beliefs, showing how alike or different they are. Although the concept of density offers a perspective on the structure of belief systems, it falls short of encapsulating the intricate architecture of a network, as it merely reflects the overall degree of connection. In this study, we examine the heterogeneity of belief systems within and between political groups using a network metric (i.e., graph diffusion distance) to quantify the heterogeneity between two networks. It is noteworthy that structural variations pertain to the difference in belief *organization* rather than the difference in the *level* of belief. Therefore, although Republicans generally have lower climate change beliefs than Democrats, the way their beliefs are organized could nonetheless be similar.

In sum, this study has three primary objectives: a) to examine the centrality of different climate beliefs, b) to analyze how the network density of climate change belief systems changes across political groups over time, and c) to compare the network structures of belief systems both within and between political groups. For data, we use fifteen waves of nationally representative cross-sectional survey data collected from 2010 to 2021. As these survey datasets are nationally representative, combining these cross-sectional surveys allows us to observe any systematic structural changes in climate change belief systems over time in the U.S.

## Methods

### Survey design and samples

This study utilized multiple waves of nationally representative cross-sectional surveys collected by the Yale Program on Climate Change Communication (YPCCC) and the George Mason University Center for Climate Change Communication (Mason 4C). The original survey datasets were collected under an exemption granted by the Institutional Review Board (IRB) of Yale University (IRB Protocol ID: 2000031972). Subsets of de-identified datasets were accessed for the purpose of this research on September 20, 2022.

After launching an initial survey in 2008, YPCCC and Mason 4C have conducted a representative survey on climate change opinions twice every year since 2010. The researchers obtained a distinct sample for each survey from Ipsos KnowledgePanel of U.S. adults aged 18 and above, recruited using probability sampling and representative of the country's population. The panel includes individuals recruited through different methods, such as random digit dialing and address-based sampling, covering nearly all U.S. residential phone numbers

and addresses. The respondents completed the survey questionnaires in a web-based environment. Individuals that did not have internet access were provided with computers and internet access.

This study utilized fifteen waves of survey data that each include fifteen common variables that were used to construct the climate change belief systems. The total number of respondents in the selected survey datasets was 16,949. We excluded 207 respondents who refused to indicate their political affiliation in the survey. These respondents are distinct from those with no party affiliation, who were retained in the sample. The former did not provide any response to the party affiliation question, while the latter explicitly said they were not affiliated with any party. Therefore, in the main analysis, there were 16,742 respondents consisting of 6,255 Republicans, 6,823 Democrats, 1,821 independent/other, and 1,843 categorized as having no party affiliation or interest in politics. The average number of respondents for each wave was 1,116 ($SD = 164.99$).

## Survey measurements

We selected fifteen variables from the survey datasets to construct network maps of the belief systems. These variables were chosen based on four criteria. First, we considered variables of significance in climate change communication research. Second, we included diverse psychological and behavioral elements, which could conceivably capture an adequate representation of a climate change belief system. Third, we endeavored to provide balance by including different types of psychological and behavioral elements within the belief system, thereby reducing any bias (e.g., an overestimated strength centrality due to strong relationships between the same type of elements). Finally, we selected variables that were included in a sufficient number of survey waves. Table 1 offers a summary of the items.

**Global warming (GW, hereafter) happening.**    To measure GW happening, we provided a short definition of global warming and then asked "Do you believe that global warming is happening?" with response options on a three-point scale: (1) "No," (2) "Don't know," and (3) "Yes" ($M = 2.49$, $SD = .77$).

**GW human cause.**    GW human cause was measured with a question asking "Assuming global warming is happening, do you think it is. . ." with response options on a four-point scale: (1) "Neither because global warming isn't happening," (2) "Caused mostly by natural changes in the environment," (3) "Caused by human activities and natural changes," and (4) "Caused mostly by human activities" ($M = 3.03$, $SD = 1.07$).

**GW consensus.**    GW consensus was measured by asking "Which comes closest to your own view?" with response options on a three-point scale: (1) "Most scientists think global warming is not happening," (2) "There is a lot of disagreement among scientists about whether or not global warming is happening," and (3) "Most scientists think global warming is happening" ($M = 2.49$, $SD = .58$).

**GW worry.**    GW worry was measured with a question asking "How worried are you about global warming?" with response options on a four-point scale: (1) "Not at all worried," (2) "Not very worried," (3) "Somewhat worried," and (4) "Very worried" ($M = 2.52$, $SD = .96$).

**Collective efficacy.**    Collective efficacy was measured with a question asking "Which of the following statements comes closest to your view?" with response options on a four-point scale: (1) "Humans can't reduce global warming, even if it is happening," (2) "Humans could reduce global warming, but people aren't willing to change their behavior, so we're not going to," (3) "Humans could reduce global warming, but it's unclear at this point whether we will do what's needed," and (4) "Humans can reduce global warming, and we are going to do so successfully" ($M = 2.45$, $SD = .85$).

Table 1. Questions and labels for questions included in the climate belief systems.

| Labels | Types | Questions |
|---|---|---|
| GW happening | Belief | Do you believe global warming is happening (Rephrased) |
| GW human cause | Belief | Assuming global warming is happening, do you think it is a human cause? |
| GW consensus | Belief | In your opinion, to what extent do scientists agree that global warming is happening (Rephrased) |
| GW worry | Other | How worried are you about global warming? |
| Collective efficacy | Other | How much do you think humans can reduce global warming? (Rephrased) |
| General attitude | Other | Do you think global warming is a bad thing or a good thing? |
| US risk | Risk perception | How much do you think global warming will harm people in the United States |
| Community risk | Risk perception | How much do you think global warming will harm your community? |
| Risk time | Risk perception | When do you think global warming will start to harm people in the United States? |
| Policy support rebate | Policy support | How much do you support or oppose the following policies? Provide tax rebates for people who purchase energy-efficient vehicles or solar panels |
| Policy support fund | Policy support | How much do you support or oppose the following policies? Fund more research into renewable energy sources, such as solar and wind power. |
| Policy support $CO_2$ | Policy support | How much do you support or oppose the following policies? Regulate carbon dioxide (the primary greenhouse gas) as a pollutant |
| Political behavior | Behavior | Over the past 12 months, how many times have you done each of the following? Written letters, emailed, or phoned government officials about global warming |
| Consumer behavior reward | Behavior | Over the past 12 months, how many times have you rewarded companies that are taking steps to reduce global warming by buying their products? |
| Consumer behavior punish | Behavior | Over the past 12 months, how many times have you punished companies that are opposing steps to reduce global warming by NOT buying their products? |

*Note*. GW = Global warming. Questions GW happening, GW consensus, and Collective efficacy are rephrased from the original survey questions in order to fit into the table. For the full text, see Measurements of Beliefs and Attitudes in the supplemental document.

**General attitude.** General attitude was measured with a question asking "On a scale from -3 (Very Bad) to +3 (Very Good) do you think global warming is a bad thing or a good thing?" with response options on a six-point scale: (1) "+3—very good," (2) "+2," (3) "+1," (4) "-1," (5) "-2," and (6) "-3" ($M = 4.63$, $SD = 1.31$).

**US risk.** US risk was measured with a question asking "How much do you think global warming will harm: people in the United States?" with response options on a four-point scale: (1) "Not at all," (2) "Only a little," (3) "A moderate amount," and (4) "A great deal" ($M = 2.66$, $SD = 1.05$).

**Community risk.** Community risk was measured with a question asking "How much do you think global warming will harm: Your community?" with response options on a four-point scale: (1) "Not at all," (2) "Only a little," (3) "A moderate amount," and (4) "A great deal" ($M = 2.42$, $SD = 1.02$).

**Risk time.** Risk time was measured with a question asking "When do you think global warming will start to harm people in the United States?" with response options on a six-point scale: (1) "Never," (2) "In 100 years," (3) "In 50 years," (4) "In 25 years," (5) "In 10 years," (6) "They are being harmed now" ($M = 3.91$, $SD = 1.93$).

**Policy support rebate.** Policy support rebate was measured with a question asking "How much do you support or oppose the following policies?: Provide tax rebates for people who

purchase energy-efficient vehicles or solar panels" with response options on a four-point scale: (1) "Strongly oppose," (2) "Somewhat oppose," (3) "Somewhat support," (4) "Strongly support" ($M = 3.06$, $SD = .89$).

**Policy support funding renewable energy.** Policy support funding renewable energy (hereafter, policy support fund) was measured with a question asking "How much do you support or oppose the following policies?: Fund more research into renewable energy sources, such as solar and wind power" with response options on a four-point scale: (1) "Strongly oppose," (2) "Somewhat oppose," (3) "Somewhat support," (4) "Strongly support" ($M = 3.13$, $SD = .88$).

**Policy support CO$_2$.** Policy support $CO_2$ was measured with a question asking "How much do you support or oppose the following policies?: Regulate carbon dioxide (the primary greenhouse gas) as a pollutant" with response options on a four-point scale: (1) "Strongly oppose," (2) "Somewhat oppose," (3) "Somewhat support," (4) "Strongly support" ($M = 2.91$, $SD = .93$).

**Political behavior.** Political behavior was measured with a question asking "Over the past 12 months, how many times have you done each of the following?: Written letters, emailed, or phoned government officials about global warming" with response options on a five-point scale: (1) "Never," (2) "Once," (3) "A few times (2–3)," (4) "Several times (4–5)," (5) "Many times (6+)" ($M = 1.23$, $SD = .71$).

**Consumer behavior reward.** Consumer behavior reward was measured with a question asking "Over the past 12 months, how many times have you done these things?: Rewarded companies that are taking steps to reduce global warming by buying their products" with response options on a five-point scale: (1) "Never," (2) "Once," (3) "A few times (2–3)," (4) "Several times (4–5)," (5) "Many times (6+)" ($M = 1.98$, $SD = 1.38$).

**Consumer behavior punish.** Consumer behavior punish was measured with a question asking "Over the past 12 months, how many times have you done these things?: Punished companies that are opposing steps to reduce global warming by NOT buying their products" with response options on a five-point scale: (1) "Never," (2) "Once," (3) "A few times (2–3)," (4) "Several times (4–5)," (5) "Many times (6+)" ($M = 1.78$, $SD = 1.30$).

**Political affiliation.** Political affiliation was used to segment respondents into four groups. Political affiliation was measured using a two-step approach. Initially, participants were asked to indicate their identification as "Republican," "Democrat," "Independent," "other," or "no party/not interested in politics." Subsequently, individuals who selected "Independent" or "other" were presented with a follow-up question inquiring whether they considered themselves closer to the "Republican party," "Democratic party," or "Neither." Respondents were classified as Republicans or Democrats if they initially identified as either party or, alternatively, did not initially identify with either party but expressed a closer affinity for one of the parties (i.e., "leaners") in the follow-up question. The "Independent" category excluded all such leaners. Participants who responded with "no party/not interested in politics" were categorized in the "no party" group.

## Network measurements

The R package *bootnet* [18] was used to estimate a climate change belief system consisting of fifteen nodes and several edges. The strength of each edge is based on a partial correlation between two nodes. Specifically, the method uses regularized partial correlation coefficients. Partial correlation coefficients offer advantages when assessing belief systems as they elucidate the relationship between two variables while controlling for influences from all other variables. Lasso regularization was used to eliminate potential spurious associations between nodes [19].

Ideally, when two variables are conditionally independent, edges should equal zero. However, partial correlation coefficients seldom reach an exact zero value. To address this limitation, the Lasso regularization technique is employed to reduce extremely weak edges to zero [18]. By discarding weak edges that potentially indicate spurious relationships, Lasso regularization prevents over-interpretation and failures to replicate estimated networks [19], as well as facilitates the estimation of a sparse belief system, which is more readily interpretable than a fully connected belief system.

The comprehensive belief system derived from all datasets was estimated in order to examine centrality of elements. Moreover, we estimated 60 belief systems, each representing a political group's belief system at each wave (4 political affiliations x 15 waves), to examine the temporal changes in density and structural heterogeneity within and between political groups.

**Centrality.**   We assessed three distinct types of centralities, including betweenness, closeness, and strength centrality. Betweenness centrality represents the degree to which a belief system component is crucial in integrating and connecting other parts of the system [7]. Closeness centrality aims to measure how quickly a particular element's influence spreads to all other parts of the belief system [7]. Strength centrality indicates an element's potential to have a greater impact on its neighboring nodes [7]. Given that the belief system in the current study is a weighted network, which is a network where the edges have assigned weights (i.e., partial correlations), we utilized methods designed to calculate centrality for weighted networks [19].

**Density.**   Density was measured by dividing the sum of absolute partial correlations by the total possible connections of a belief system. Theoretically, density ranges from 0, indicating no edges, to 1, indicating a fully connected network where every edge has a weight of 1 ($M = .06$, $SD = .01$).

**Structural difference.**   We employed the Graph Diffusion Distance (GDD) metric to measure the structural difference between two belief systems, using the R package *NetworkDistance* [20]. This measurement operates under the assumption that if two networks sharing the same set of nodes possess distinct structures, their diffusion patterns of information or something that flows through these networks will vary [21]. By simulating and comparing these diffusion patterns, the algorithm quantifies the dissimilarity between two compared networks. Theoretically, GDD can range from 0, denoting identical networks, to infinity. To constrain the values within a range of 0 to 1, we normalized GDD using the min-max method. GDD values were calculated for each pair within the sixty belief systems, resulting in a total of 1,770 GDD values. These comprised 420 GDD values for intra-group comparisons ($M = .36$, $SD = .12$) and 1,350 GDD values for inter-group comparisons ($M = .41$, $SD = .12$).

## Data analysis

All statistical analyses were conducted using R, version 4.3.1 [22]. In a belief system, there is only a single centrality score per element, which prevents us from evaluating the score statistically. To address this limitation, we employed a bootstrap method. As a non-parametric sampling technique, it allows for estimating the distribution of statistical parameters such as confidence intervals without making assumptions about the form of the underlying population distribution. The implemented bootstrap repeatedly sampled 1,000 respondents with replacement from the observed data and estimated the 95% confidence interval of the centrality score for each element. To assess density change over time and across different political groups, we conducted a regression analysis in which density is regressed on an indicator for survey wave, political affiliations, and their interaction terms. A simple slope analysis is conducted to further clarify the strength and direction of the density trends within the political affiliation. To assess

the structural differences (GDD) both within and between political groups, we carried out pairwise comparisons using Tukey's Honestly Significant Difference (HSD) test.

## Results

Fig 1 illustrates climate change belief systems for all Americans and further breaks them down by political group. Fig 2 demonstrates the three types of centrality scores for each belief element. Here, we report mean centrality and its 95% confidence interval based on the bootstrap.

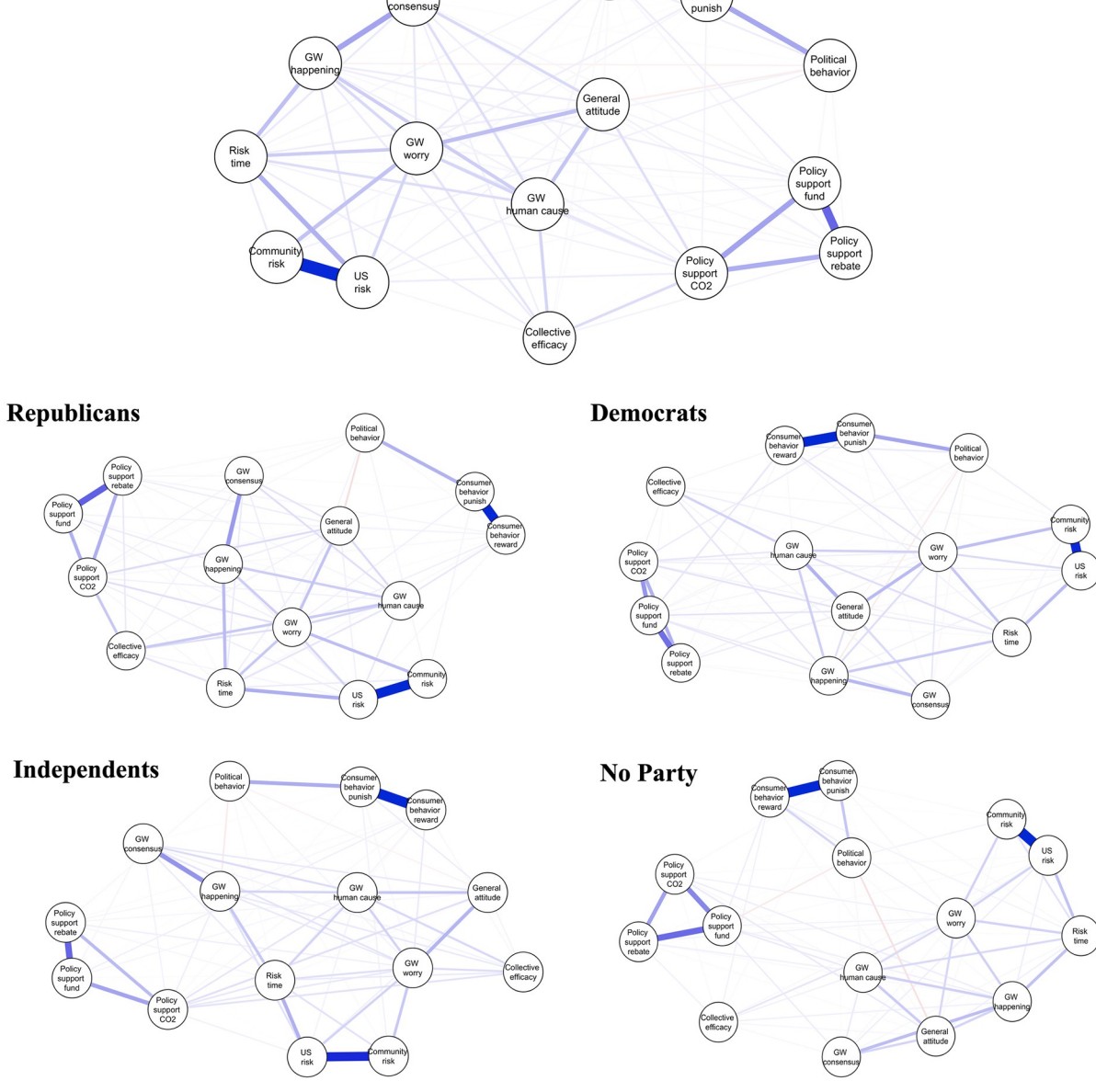

**Fig 1. Climate change belief system of all, Republicans, Democrats, Independents, and no party.** *Note*. Questions for all variables are shown in Table 1. The color of an edge represents the sign of the partial correlation, with blue indicating a positive partial correlation and red indicating a negative partial correlation. The thickness of an edge represents the strength of the partial correlation. The thicker edge indicates a stronger partial correlation.

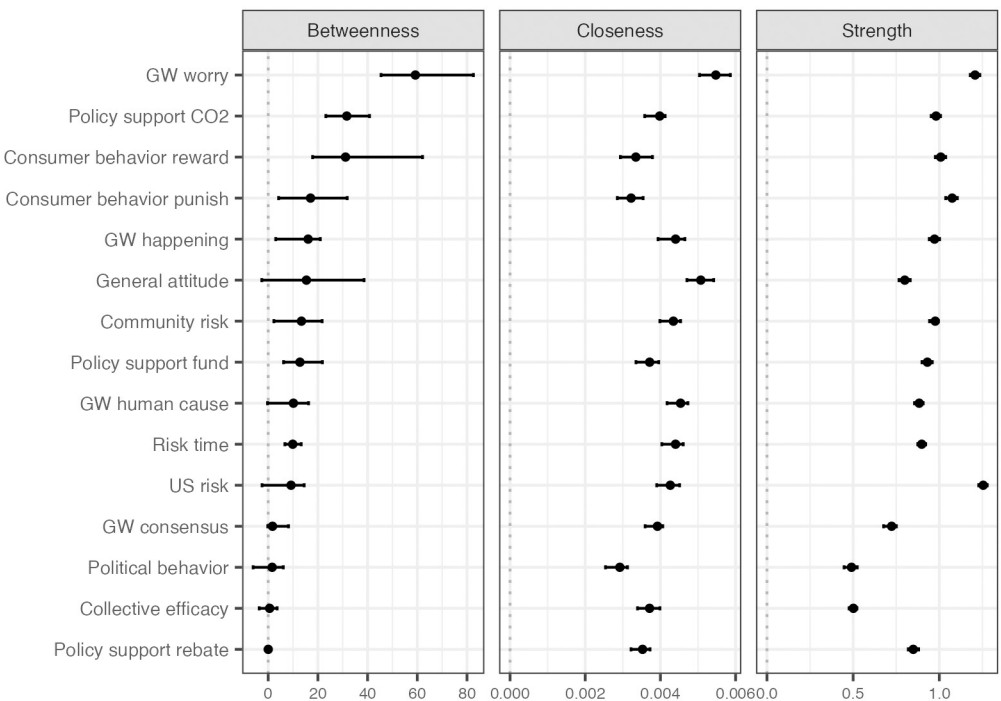

**Fig 2. Betweenness, closeness, and strength centralities for each element.** *Note.* Variables are ordered by betweenness centrality score. Centrality scores were estimated from the belief system from all participants. Each point on the graph represents the mean centrality score for a specific belief element, and error bars represent the 95% confidence interval for each estimate.

Worry about global warming (hereafter "worry") is the most central element of the belief system. The highest betweenness centrality score for worry indicates that it plays a significant role in connecting disparate elements such as beliefs, attitudes, policy support, and behaviors (μ = 59.23, 95% CI [45.42, 82.58]). Moreover, worry also has the highest closeness centrality score, indicating that changes in worry can influence other psychological elements or that changes in other elements can in turn influence worry (μ = .005, 95% CI [.005,.006]).

Worry has the second highest strength centrality score, indicating that it has the potential to substantially impact its neighboring elements (μ = 1.21, 95% CI [1.18, 1.24]). Although US risk was estimated to have the highest strength centrality score (μ = 1.25, 95% CI [1.23, 1.28]), a substantial variance in this score (73%) was driven largely by its strong relationships with community risk perception (edge strength between US risk and community risk = .69, 95% CI [.68,.70]) and risk time (edge strength between US risk and risk time = .22, 95% CI [.20,.23]), indicating that changes in the US risk perception will likely influence only these neighboring elements or vice versa. Moreover, the lower scores for US risk in both betweenness (μ = 9.18, 95% CI [-2.48, 14.48]) and closeness centrality (μ = .004, 95% CI [.003,.004]) compared to those for worry further lends further evidence to our conclusion that worry is the most central element in the climate change belief system.

The pairwise comparison between worry and other elements indicates that the centrality scores of worry, including betweenness, closeness, and strength, are significantly higher than those of other elements, except for the strength centrality of US risk. The detailed results of the comparison are available in S1–S3 Tables in the supplemental material.

Next, we examined the density of belief systems across political groups over the past decade. Fig 3 illustrates the results. In the analysis, we used Republicans as a reference, meaning that

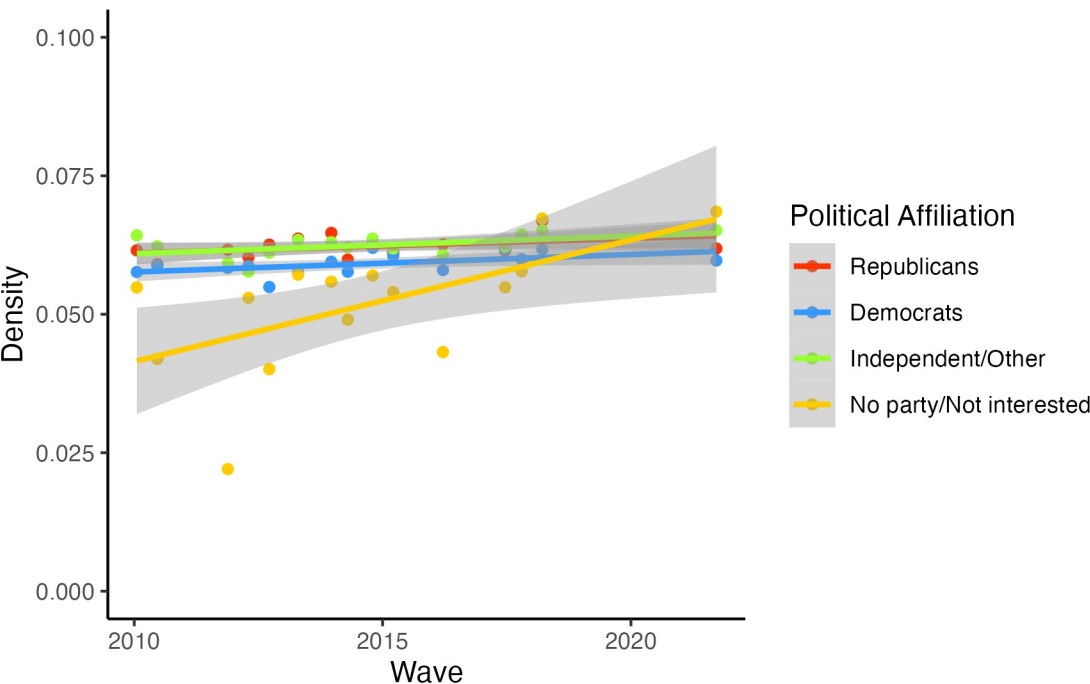

**Fig 3. Changes in belief system density across political affiliations for the past decade.**

the density of each group was compared with that of Republicans. It is worth noting that the choice of a reference group does not influence or alter the outcomes of the analysis. While the group with no political affiliation generally had less dense belief systems compared to Republicans ($b = -.02$, $p <. 001$, 95% CI [-.03, -.01]), the density of the no affiliation group increased significantly over time relative to Republicans ($b = .001$, $p < .01$, 95% CI [.001,.002]). The results of the simple slope analysis further reveals a significant increase in the density of the no affiliation group (*estimated slope b* = .002, 95% CI [.001,.002]), whereas the density of groups with political affiliations remained relatively stable over time (Republicans: $b = .0002$, 95% CI [-.0004,.001]; Democrats: $b = .0003$, 95% CI [-.0003,.001]; Independent/Other: $b = .0002$, 95% CI [-.0004,.001]).

We compared the overall structure of belief systems to evaluate the extent to which political groups develop homogeneous or heterogeneous belief systems both within and between groups. Fig 4 illustrates the results. First, intra-group comparisons of belief systems across multiple time points revealed that the politically unaffiliated developed more heterogeneous belief systems within the group ($M = .49$, $SD = .14$) compared to other groups, including Democrats ($M = .27$, $SD = .03$; *difference* (*diff*, hereafter) = .21, $p < .001$), Republicans ($M = .27$, $SD = .04$, *diff* = .21, $p < .001$), and Independents ($M = .39$, $SD = .04$, *diff* = .09, $p < .001$). Republicans developed relatively more homogenous belief systems within their group than Independents (*diff* = .12, $p < .001$). In addition, Democrats also developed more homogenous belief systems than did Independents (*diff* = .12, $p < .001$). There was no significant difference in GDD score between Republicans and Democrats, indicating that the level of intra-homogeneity was similar between the two groups (*diff* = .001, $p = 1.00$).

We also conducted inter-group comparisons to compare the overall structure of belief systems between different political groups. The politically unaffiliated had more heterogeneous belief systems compared to other political groups. Specifically, the unaffiliated shared the least

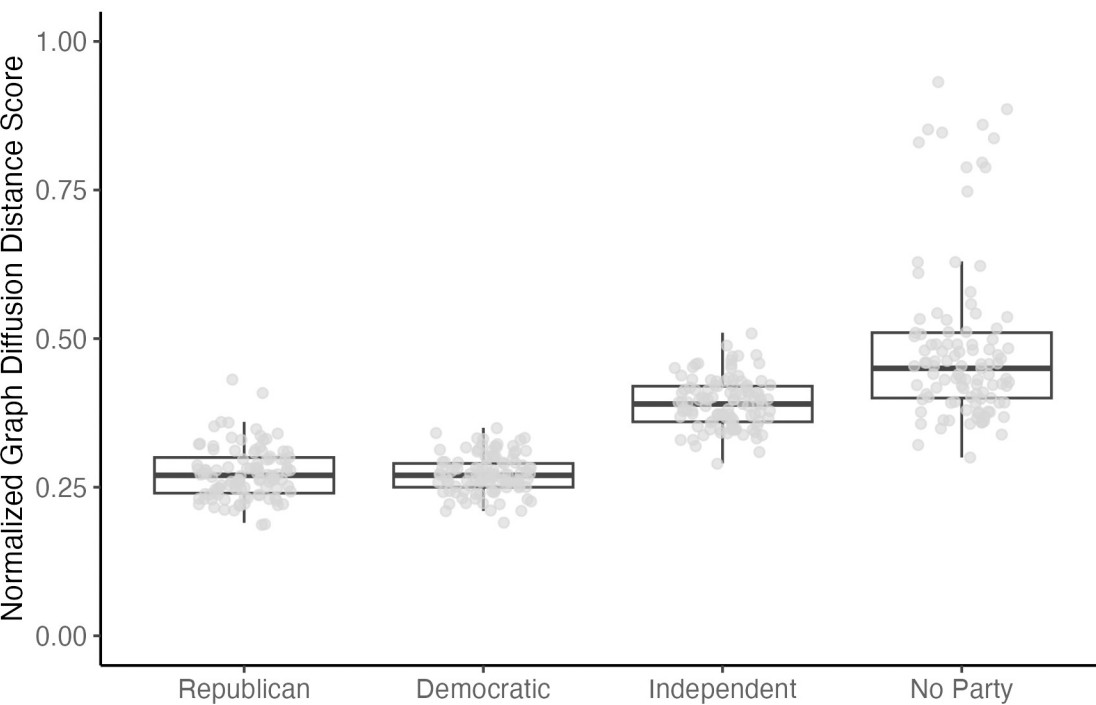

**Fig 4. Intra-group comparison of belief systems for each political group.** *Note.* Each dot represents the graph diffusion distance (GDD) score of two belief systems. More specifically, each dot represents a comparison between different waves of data collected at different time points, within each political group. GDD scores are normalized to fall within 0 to 1.

overlapping structure with the Independents ($M$ = .50, $SD$ = 13), followed by the Republicans ($M$ = .47, $SD$ = 13) and the Democrats ($M$ = .44, $SD$ = 13). Comparing the belief systems of the three groups indicated that the average GDD score for the pair of Republicans and Democrats ($M$ = .34, $SD$ = .05) was significantly lower than that of the pair of Democrats and Independents ($M$ = .37, $SD$ = .04), diff = .03, p < .01. In other words, Democrats had a more structurally similar belief system to Republicans than to Independents. There was no significant difference between the pair of Republicans and Democrats ($M$ = .34, $SD$ = .05) and the pair of Republicans and Independents ($M$ = .35, $SD$ = .04), *diff* = .01, *p* = .96. In other words, the level of heterogeneity was similar between the pair of Republicans and Democrats and the pair of the Republicans and Independents. These results are illustrated in Fig 5. We performed an outlier sensitivity test to ensure the findings were robust without outliers. The results showed that the findings were nearly identical before and after removing outliers. The details of the sensitivity test can be found in the supplementary material.

## Discussion

This study indicates that worry about global warming is the most central element of climate change belief systems in the United States. Although other studies have also identified worry as a significant factor [1, 23], our investigation reaffirms its importance from a structural standpoint, finding that worry plays multiple important roles in people's climate change belief systems. Worry plays a significant role in connecting various psychological elements, including beliefs, risk perceptions, attitudes, policy support, and behaviors. Moreover, changes in the level of worry have a greater potential than other elements to have cascading impacts on other psychological and behavioral elements within the system.

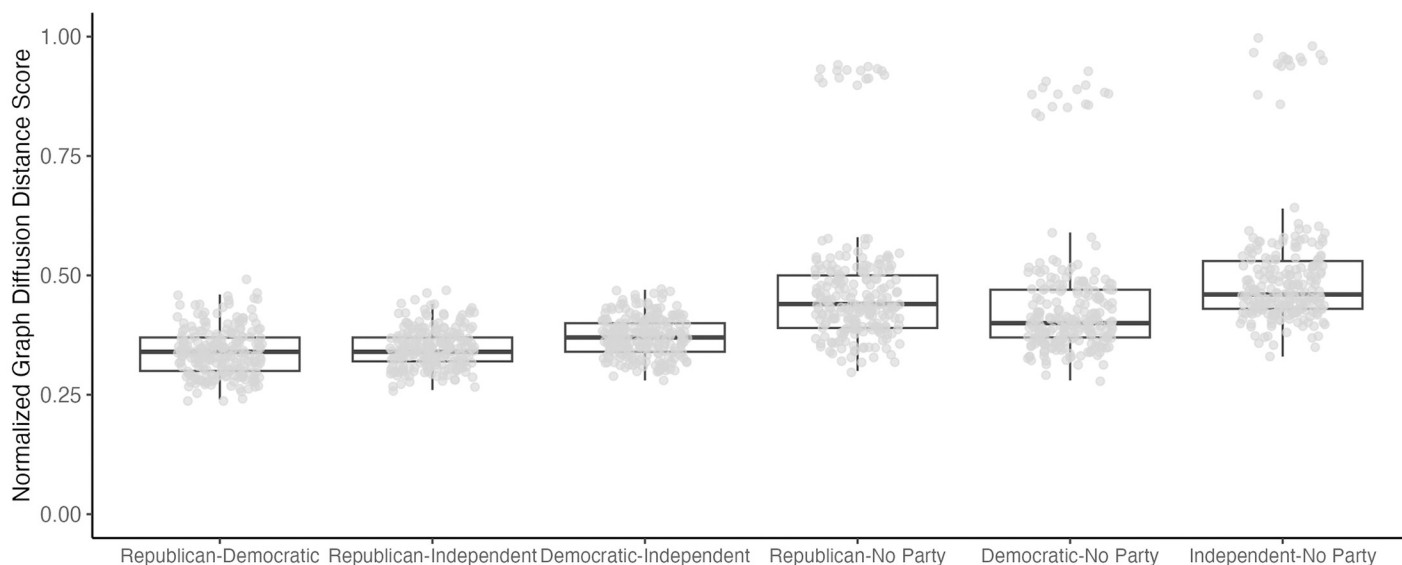

**Fig 5. Inter-group comparison of belief systems between political groups.** *Note.* Each dot represents the graph diffusion distance (GDD) score of two belief systems. More specifically, each dot represents a comparison between different or same waves of data collected at different or same time points, between two political groups. GDD scores are normalized to fall within 0 to 1.

The significant and multi-faceted roles of worry, as identified in this study, could aid climate change communicators in devising more precise communication strategies. As worry serves as a critical psychological bridge, it can facilitate shifts in beliefs and attitudes from one element to others within the system. This finding aligns with prior research indicating that worry might be a pivotal mediator, potentially forging links between beliefs and their consequent outcomes [3, 24]. Given its proximal position and potent ties with other elements, activating worry should trigger a cascade of changes in other psychological components. Worry is not only positively associated with support for taking action [3], but can also encourage people to engage with the issue on a personal level [25]. This underscores that worry is a vital psychological construct that merits emphasis in climate change communication, a conclusion supported by both psychological evidence and our own observations from a structural perspective.

Despite the central role of worry, this research does not imply that worry is a universal factor that every message should focus on. Addressing worry might be effective when the intention of communication is to achieve a broad impact on the entire belief system. However, when the aim is more specific, focusing directly on other psychological or behavioral elements may be more effective. For example, influencing consumer activism may offer a more direct path to influencing political actions, given the tight link between these elements in the climate change belief system. Moreover, it is worth noting that it may be more challenging to influence worry than other peripheral elements because central elements are more tightly interwoven with other beliefs and attitudes, making them harder to change [12].

Researchers should also exercise caution when determining the appropriate level of worry to induce. Excessive worry can lead individuals to engage in maladaptive behaviors such as avoidance, rather than problem-solving, particularly when efficacy to respond to the issue is low [26]. A pertinent example of this is the rising phenomenon of "eco-anxiety," or potentially debilitating chronic worry about the environmental crisis. For some, the overwhelming scale of climate change can elicit extreme worry, which, instead of prompting constructive action, leads to feelings of despair. Therefore, it is also important to provide efficacy messages, which

can empower individuals with the agency to effect change, and help prevent worry from morphing into paralyzing anxiety. Future research needs to investigate the role of efficacy messages in fostering a positive influence of worry within belief systems.

Our research also reveals significant changes in the density of climate change belief systems over the past decade, predominantly among individuals without political affiliations. Specifically, people without political affiliations have experienced a significant increase in density, in contrast to politically affiliated groups, which have largely maintained steady density levels. This pattern is echoed in intra-group comparisons. The increase in density, particularly among politically unaffiliated individuals, represents heterogeneities of belief systems within this group across time, contrasting with the relative unchanging structures amongst their politically affiliated peers over the same period. The findings about density align with previous studies suggesting that those less engaged with U.S. politics have less tightly organized belief systems [9]. Our research adds a new dimension to this finding, indicating that those unaffiliated with a political party have experienced an increase in the organization of their belief systems over time. This development could be influenced by the greater public discourse, expanded awareness, and changes in attitudes toward climate change that have occurred over the past decade [6]. Yet, it remains uncertain which factors specifically contributed to this change in density among politically unaffiliated individuals. Future research is needed to answer this intriguing question.

Interestingly, the inter-group comparisons demonstrate a greater level of structural homogeneity in belief systems between Republicans and Democrats compared to other group pairings, with the exception of the pair comparing Republicans and Independents. This suggests that despite Republicans' lower levels of pro-climate beliefs and attitudes in comparison to Democrats [6], the underlying *structure* that governs how individuals from both groups organize their belief systems is not markedly different. This finding implies that bolstering core pro-climate beliefs and attitudes among Republicans could lead to subsequent support for climate change policies and actions, similar to their Democratic counterparts. Indeed, previous findings support this notion by suggesting that effective climate change communication can increase policy support among Republicans [27].

## Limitations and future research direction

The current study has limitations. First, this study uses cross-sectional survey datasets, not controlled experiments, thus cannot provide causal explanations. This may obscure the causal implications of central belief elements. For instance, it remains unclear whether worry, identified as the central element, is the cause or the consequence of other elements such as risk perceptions. Determining the causes of change in a belief system is challenging, as it requires numerous manipulations and strong assumptions. Nevertheless, theories and empirical evidence from experimental research suggest that worry is both a cause and consequence [3] as well as a strong predictor of environmental policy support and behaviors [1, 23]. Therefore, it is important to interpret this study in the context of previous studies to enrich our understanding of where climate change beliefs and attitudes fit within a broader psychological system and what the causal directions are.

Second, there is a need for future research on effective ways to cultivate a productive level of worry about global warming through communication. Previous research has shown that emphasizing the scientific consensus on climate change does increase worry, but only to a modest degree [3]. An alternative and potentially more effective strategy is 'worry modeling,' in which trusted figures explicitly state their concerns about climate change and explain the basis for their worry. This approach holds promise based on social cognitive theory and social

norm theory. First, role modeling can help individuals learn and adopt similar emotional states from a trusted figure [28]. Second, role modeling can help establish a perceived social norm that worrying about climate change is common, proper, and even imperative. The impact of worry modeling could be further amplified by utilizing multimodal content that includes non-verbal cues, such as facial expressions and tone of voice, providing immediate indicators of emotional urgency [29]. We call for future research to investigate the potential of this intriguing strategy.

Estimating belief systems at the aggregate level presents another caveat. While using nationally representative survey data allows us to approximate the belief system of an average individual within a given group [30], it does not necessarily follow that all individuals in that group share the same belief system structure. Promisingly, recent studies have attempted to estimate belief systems at the individual level [31]. Future research could try to match belief system analysis at multiple levels to identify similarities and differences.

The findings of this study should also not necessarily be generalized to other countries. Although climate change is a global issue, public beliefs and attitudes toward climate change vary widely by country [32]. The U.S. likely has substantially different climate change belief systems than other countries in this regard. Given the tremendous variation in public responses to climate change around the world, climate change beliefs systems are also likely to vary. Belief systems are presumably affected by many factors such as culture, education, and other socio-economic factors. As the belief system framework can provide valuable insights for the development of strategic climate change communication, implementing the belief system framework on a global scale is encouraged. This could shed light on the diversity of belief systems around the world and help us develop more effective communication strategies that take into account the context of each country.

Overall, the findings demonstrate that the belief system approach is a valuable framework that enriches our understanding of the organization and interplay of different climate change beliefs. Worry about climate change is identified as the central element in American climate change belief systems. Over the past decade, the organization of belief systems among politically disengaged individuals has increased, implying that improved climate change communication is helping people develop more coherent and cohesive belief systems on the subject. Despite the pronounced discrepancy in the strength of beliefs between the political left and right, our findings reveal a structural similarity in how belief systems are organized within these two groups. Collectively, these insights can guide the development of effective climate change communication strategies.

## Supporting information

**S1 Fig. Centrality stability.**
(TIF)

**S2 Fig. Intra-group comparison of belief system after removing outliers.**
(TIF)

**S3 Fig. Inter-group comparison of belief systems after removing outliers.**
(TIF)

**S1 Table. Betweenness centrality difference between worry and other elements.**
(DOCX)

**S2 Table. Closeness centrality difference between worry and other elements.**
(DOCX)

**S3 Table. Strength centrality difference between worry and other elements.**
(DOCX)

**S4 Table. Difference of GDD scores in intra-group comparison of belief systems.**
(DOCX)

**S5 Table. Difference of GDD scores in inter-group comparison of belief systems.**
(DOCX)

## Author Contributions

**Conceptualization:** Sanguk Lee, Matthew H. Goldberg.

**Data curation:** Seth A. Rosenthal.

**Formal analysis:** Sanguk Lee.

**Methodology:** Sanguk Lee.

**Resources:** Seth A. Rosenthal, Edward W. Maibach, John E. Kotcher, Anthony Leiserowitz.

**Writing – original draft:** Sanguk Lee, Matthew H. Goldberg.

**Writing – review & editing:** Sanguk Lee, Matthew H. Goldberg, Seth A. Rosenthal, Edward W. Maibach, John E. Kotcher, Anthony Leiserowitz.

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
