## [Decision Letter · Decision Letter 0]

8 Oct 2023

PONE-D-23-28322Climate Change Belief Systems across Political Groups in the United StatesPLOS ONE

Dear Dr. Lee,

Thank you for submitting your manuscript to PLOS ONE. After careful consideration, we feel that it has merit but does not fully meet PLOS ONE’s publication criteria as it currently stands. Therefore, we invite you to submit a revised version of the manuscript that addresses the points raised during the review process.

We look forward to receiving your revised manuscript.

Kind regards,

Enkeleint A. Mechili

Academic Editor

PLOS ONE

5. We notice that your supplementary figures are uploaded with the file type 'Figure'. Please amend the file type to 'Supporting Information'. Please ensure that each Supporting Information file has a legend listed in the manuscript after the references list.

Reviewers' comments:

Reviewer's Responses to Questions

**Comments to the Author**

1. Is the manuscript technically sound, and do the data support the conclusions?

Reviewer #1: Yes

Reviewer #2: Partly

2. Has the statistical analysis been performed appropriately and rigorously? 

Reviewer #1: Yes

Reviewer #2: Yes

3. Have the authors made all data underlying the findings in their manuscript fully available?

Reviewer #1: Yes

Reviewer #2: Yes

4. Is the manuscript presented in an intelligible fashion and written in standard English?

Reviewer #1: Yes

Reviewer #2: Yes

5. Review Comments to the Author

Reviewer #1: Title: Climate Change Belief Systems across Political Groups in the United States

Abstract: Key study aim, methods and result of the study well presented.

Introduction: Detailed information on statement of problem, rational for the study clearly presented and study objective well presented.

Methods: Well described.

• Please clarify the difference between the 207 respondents who did not indicate political affiliation and the 1,843 categorized as having no party affiliation.

Result: Well written in details with relevant figures

Discussion: The study findings are well discussed, with study limitations provided.

Conclusion: Clearly written with appropriate recommendation.

Reviewer #2: 1. The manuscript lacks technicalities, but the data reflects the Conclusion.

2. Information on the statistical analysis, including the statistical tool used is sketchy; and needs to be elaborated. For instance, there should be a subheading for Data Analysis. There is need to mention the version of the statistical tool used, including manufacturer, city and country manufactured.

3. Nil concerns.

4. The study rationale is not properly captured in the 'Introduction'. The identified knowledge gap(s) should be adequately rationalised to maintain a logical flow in the ‘Introduction’.

6. PLOS authors have the option to publish the peer review history of their article (what does this mean?). If published, this will include your full peer review and any attached files.

Reviewer #1: **Yes: **Tanimola Makanjuola Akande

Reviewer #2: No

---

## [Author Response · Author response to Decision Letter 0]

4 Jan 2024

Authors’ Response to Reviewer #1

We appreciate the reviewer's positive feedback throughout the manuscript. One concern the reviewer highlighted pertains to the distinction between the 207 respondents who did not indicate a political affiliation and the 1,843 categorized as having no party affiliation. We are grateful to the reviewer for pointing out this ambiguity. These groups differ in that the former chose not to disclose their political affiliation by not answering the question, while the latter explicitly stated they were not affiliated with any political party. We have clarified this distinction in the "Survey Design and Samples" section of the manuscript (see revised lines 301 and 304 on page 7). We believe these changes will help readers understand the distinction between the two groups more clearly.

Authors’ Response to Reviewer #2

We appreciate the reviewer's constructive feedback throughout the paper. Please find our itemized responses below. Our responses are italicized.

1. The manuscript lacks technicalities, but the data reflects the Conclusion.

[Authors’ Response] We appreciate the reviewer’s feedback. To bolster the technical depth of the paper, we have taken into account the suggestions provided in the reviewer’s second comment. Our actions to enhance the manuscript's technicalities are detailed in our response to the second comment provided by the reviewer.

2. Information on the statistical analysis, including the statistical tool used is sketchy; and needs to be elaborated. For instance, there should be a subheading for Data Analysis. There is need to mention the version of the statistical tool used, including manufacturer, city and country manufactured.

[Authors’ Response] We appreciate the reviewer’s constructive comments that enhance the quality of the paper. We concur with the reviewer's observation that the original manuscript lacked detail in explaining the analysis and the statistical tool used. As the reviewer suggested, we have added a new section titled “Data Analysis” to provide details about the version of our statistical tool (R 4.3.1) and our data analysis approach (see revised lines 451 to 464 on page 14). Furthermore, we have now included a reference for our statistical tool (R Core Team, 2022), ensuring readers are informed about the manufacturer, city, and country of manufacture.

Reference

R Core Team. R: A language and environment for statistical computing [Internet]. Vienna, Austria: R Foundation for Statistical Computing; 2022. Available from: https://www.R-project.org/

3. The study rationale is not properly captured in the 'Introduction'. The identified knowledge gap(s) should be adequately rationalised to maintain a logical flow in the ‘Introduction’.

[Authors’ Response] We are grateful for the reviewer's feedback, which has helped us to refine the rationale of our study. In response to the suggestion that the identified knowledge gaps require clearer justification, we have increased the detail of our literature review (see revised lines 39 to 42 on page 3). This includes the addition of a study (Verschoor et al., 2020) that applies the belief system framework to the context of climate change, which we have now referenced to bolster the rationale for our research. Moreover, we have added the third paragraph of introduction to better articulate the focus of our study and the significance of our investigation (see revised lines 43 to 66 on page 3 and 4).

Reference

Verschoor M, Albers C, Poortinga W, Böhm G, Steg L. Exploring relationships between climate change beliefs and energy preferences: A network analysis of the European Social Survey. Journal of Environmental Psychology. 2020 Aug;70:101435.

---

## [Decision Letter · Decision Letter 1]

22 Feb 2024

Climate Change Belief Systems across Political Groups in the United States

PONE-D-23-28322R1

Dear Dr. Lee,

We’re pleased to inform you that your manuscript has been judged scientifically suitable for publication and will be formally accepted for publication once it meets all outstanding technical requirements.

Kind regards,

Enkeleint A. Mechili

Academic Editor

PLOS ONE

Additional Editor Comments (optional):

Reviewers' comments:

Reviewer's Responses to Questions

**Comments to the Author**

1. If the authors have adequately addressed your comments raised in a previous round of review and you feel that this manuscript is now acceptable for publication, you may indicate that here to bypass the “Comments to the Author” section, enter your conflict of interest statement in the “Confidential to Editor” section, and submit your "Accept" recommendation.

Reviewer #3: All comments have been addressed

Reviewer #4: All comments have been addressed

2. Is the manuscript technically sound, and do the data support the conclusions?

Reviewer #3: Yes

Reviewer #4: Yes

3. Has the statistical analysis been performed appropriately and rigorously? 

Reviewer #3: Yes

Reviewer #4: Yes

4. Have the authors made all data underlying the findings in their manuscript fully available?

Reviewer #3: Yes

Reviewer #4: Yes

5. Is the manuscript presented in an intelligible fashion and written in standard English?

Reviewer #3: Yes

Reviewer #4: Yes

6. Review Comments to the Author

Reviewer #3: The manuscript is good but you have to focus on the interpretation of the data also mention the data collection methods in brief i.e, why you selected the specific numbers respondent?

Reviewer #4: Dear Authors,

Your manuscript comprehensively analyzes climate change belief systems, effectively utilizing bootstrap methods, regression analysis, and the novel application of Graph Diffusion Distance (GDD) to measure belief system homogeneity. The findings on the centrality of worry in these systems and its potential role in climate communication are particularly insightful. However, I recommend further elaboration on your methodological choices, such as the rationale behind the number of bootstrap samples and a more detailed explanation of GDD for clarity. Additionally, balancing the emphasis on worry with considerations of potential maladaptive behaviors due to excessive concern would strengthen your discussion. Your exploration of belief system dynamics across political affiliations offers valuable insights, but future research could benefit from a longitudinal or experimental approach to better establish causal relationships. Overall, your study contributes significantly to understanding the interplay between psychological factors and climate change beliefs, and its implications for climate change communication are particularly noteworthy. I appreciate your work, and I recommend your work to be accepted to published and inform our global readers and scholars.

Thank you,

A Reviewer

7. PLOS authors have the option to publish the peer review history of their article (what does this mean?). If published, this will include your full peer review and any attached files.

Reviewer #3: **Yes: **Dr. Najaf Ali Shigri

Reviewer #4: **Yes: **Muhammad Hassan Bin Afzal

---

## [Editor Report · Acceptance letter]

27 Feb 2024

PONE-D-23-28322R1 

PLOS ONE

Dear Dr. Lee, 

I'm pleased to inform you that your manuscript has been deemed suitable for publication in PLOS ONE. Congratulations! Your manuscript is now being handed over to our production team.

Kind regards, 

on behalf of

Prof. Assoc. Dr. Enkeleint A. Mechili 

Academic Editor

PLOS ONE